# The Research Progress of Mitochondrial Transplantation in the Treatment of Mitochondrial Defective Diseases

**DOI:** 10.3390/ijms25021175

**Published:** 2024-01-18

**Authors:** Cuilan Hu, Zheng Shi, Xiongxiong Liu, Chao Sun

**Affiliations:** 1Institute of Modern Physics, Chinese Academy of Sciences, Lanzhou 730000, China; hucuilan23@mails.ucas.ac.cn (C.H.); sz0326@impcas.ac.cn (Z.S.); lxx002@impcas.ac.cn (X.L.); 2Key Laboratory of Heavy Ion Radiation Biology and Medicine, Chinese Academy of Sciences, Lanzhou 730000, China; 3Key Laboratory of Basic Research on Heavy Ion Radiation Application in Medicine, Lanzhou 730000, China; 4University of Chinese Academy of Sciences, Beijing 100049, China

**Keywords:** mitochondrial transplantation, mitochondrial defective diseases, cerebral ischemia, myocardial ischemia-reperfusion, cancer

## Abstract

Mitochondria are double-membrane organelles that are involved in energy production, apoptosis, and signaling in eukaryotic cells. Several studies conducted over the past decades have correlated mitochondrial dysfunction with various diseases, including cerebral ischemia, myocardial ischemia-reperfusion, and cancer. Mitochondrial transplantation entails importing intact mitochondria from healthy tissues into diseased tissues with damaged mitochondria to rescue the injured cells. In this review, the different mitochondrial transplantation techniques and their clinical applications have been discussed. In addition, the challenges and future directions pertaining to mitochondrial transplantation and its potential in the treatment of diseases with defective mitochondria have been summarized.

## 1. Introduction

Mitochondria are the “powerhouses” of eukaryotic cells and the primary sites of intracellular adenosine triphosphate (ATP) production through oxidative phosphorylation and tricarboxylic acid cycle [1]. In addition, the mitochondria are also involved in the regulation of cell growth and differentiation, cell signaling, apoptosis, and cell cycle [2].

Mitochondria are susceptible to damage from various stress stimuli, including ischemia, hypoxia, radiation, and drugs [3]. The dysfunctional mitochondria produce excessive levels of reactive oxygen species (ROS), and the resulting oxidative damage to macromolecules and cells can eventually lead to abnormal organ function, systemic disease, or even death [4].

Mitochondrial transplantation involves the transfer of healthy mitochondria into cells or tissues with dysfunctional mitochondria [5]. The healthy mitochondria can be derived from autologous or allogeneic sources, such as skeletal muscle cells, mesenchymal stem cells (MCSs), or cell lines. The transplanted organelles are taken up by the recipient cells and integrated into the existing mitochondrial network, thereby enhancing mitochondrial function and promoting cellular viability [6].

Recent studies have shown that mitochondria can be transferred between similar and different types of somatic cells, a phenomenon known as horizontal mitochondrial transfer (HMT) [7]. HMT has been observed both in vitro and in vivo, as well as under physiological and pathological conditions. It also occurs during tumor progression and lung injuries, where it plays a role in bio-energetic signaling, maintaining homeostasis, recovery, and therapeutic response and recalcitrance [8,9]. In addition to intercellular transfer, mitochondria also undergo constant morphological changes through fusion and fission [10].

## 2. Methods of Mitochondrial Transplantation

### 2.1. Natural Mitochondrial Transfer Mechanisms

Mitochondria are highly dynamic organelles that continuously change their shape through fusion and fission, forming a network structure [11]. In recent years, the intercellular transfer of mitochondria has become increasingly important as it promotes the integration of transferred mitochondria into the endogenous network of recipient cells, leading to changes in the recipient cells’ bioenergetic features and other functional characteristics [12]. Several natural mechanisms have been identified for this transfer, including tunneling nanotubes (TNTs), extracellular vesicles (EVs), and gap junction channels (GJCs) [13] (Figure 1).

TNTs are F-actin-based transient filamentous membrane protrusions or cytoplasmic bridges that allow cell-to-cell communication and the transfer of biological material [14]. In addition, the TNTs also serve as conduits for the trafficking of mitochondria and other organelles, such as endosomes, ER, Golgi/ER, lysosomes, and melanosomes [15,16]. HMT, a major function of the TNTs, protects cells against mitochondrial injury in both physiological and pathological states [17]. Ahmad et al. showed that MSCs can form TNTs with epithelial cells in response to various stimuli (e.g., rotenone (100 nM) or TNF-α (20 ng)) and transfer mitochondria to the latter via these nanotubes, thus restoring mitochondrial function in the stressed recipient cells and aiding in their repair and regeneration [18]. Wang et al. discovered that co-culturing UV-treated pheochromocytoma (PC) 12 cells with untreated PC12 cells led to the formation of various types of TNTs, characterized by the presence of continuous microtubules inside these TNTs, forming microtubule-containing TNTs (MT-TNTs). Within these TNTs, mitochondria exhibited co-localization with microtubules and were transported along these structures from healthy cells to stressed cells. The MT-mediated functional transfer of mitochondria during the early stages of apoptosis exerted a reversal effect on stressed cells, ultimately rescuing the damaged cells. These newly observed TNTs play a crucial role as determinants for long-distance organelle transport [19].

Cell fusion is a mechanism analogous to the formation of TNTs, whereby cytoplasmic constituents and organelles are equitably shared between adjacent cells of distinct cell types. This process occurs through physical connections of their plasma membranes, leading to their permanent fusion [20,21]. Acquistapace et al. conducted a co-culture experiment involving fully differentiated mouse cardiomyocytes and human multipotent adipose-derived stem cells, wherein they observed the partial fusion of heterologous cells. Notably, the transfer of mitochondria from stem cells to cardiomyocytes was observed, and the involvement of intercellular structures, specifically those composed of F-actin and microtubules, was revealed. Furthermore, it was shown that adult stem cells are capable of reprogramming cardiomyocytes to a progenitor-like state through the process of partial cell fusion and subsequent transfer of functional mitochondria [22].

Mitochondria are also transported from donor cells via EVs [23] and enter the cytoplasm of the recipient cells through endocytosis or fusion [24]. Zhang et al. found that EV-mediated mitochondrial transfer from human umbilical cord-derived mesenchymal stem cells to okadaic acid-treated SH-SY5Y cells, an in vitro model of Alzheimer’s disease, reversed apoptotic signaling, improved cellular immune response, and reduced oxidative stress [25]. Furthermore, Ikeda et al. discovered that in an in vivo model of myocardial infarction in mice, functional mitochondrial vesicles rich in induced pluripotent stem cell-derived cardiomyocytes (iCMs) can be isolated from the centrifugation of iCM culture medium. These mitochondria-rich EVs (M-EVs) transfer their mitochondrial cargo to hypoxia-injured iCMs, integrating into the endogenous mitochondrial network. Treatment with 1.0 × 10^8^/mL M-EVs significantly impacts ATP production, restoring cellular energy and mitochondrial biogenesis in cardiomyocytes, thus enhancing cardiac function after myocardial infarction [24]. These findings confirmed that mitochondria can enter target cells through EVs.

Gap junctions are specialized intercellular channels that enable direct communication between adjacent cells. These channels allow the passage of ions, metabolites, and signaling molecules, thereby facilitating coordinated responses in tissues and organs [26]. Recent studies have shown that gap junctions also serve as conduits for mitochondrial exchange between cells and, therefore, play a crucial role in diseases associated with mitochondrial dysfunction [27]. Three-dimensional electron microscopy and immunogold labeling of the gap junction protein connexin 43 (Cx43) have shown that entire organelles, including mitochondria and endosomes, are incorporated into double-membrane vesicles, known as connexosomes or annular gap junctions, that form as a result of gap junction internalization [28,29]. Li et al. co-cultured bone marrow-derived mesenchymal stem cells (BMSCs) with VSC4.1 motor neurons subjected to oxygen–glucose deprivation (OGD), a cellular model of spinal cord injury (SCI), and found that mitochondria transferred from the BMSCs to the injured neurons via gap junctions promoted neuronal survival by preventing OGD-induced apoptosis and restored motor function. The heterotypic gap junctions between BMSCs and neurons are formed by connexin 43 and connexin 32, and mitochondrial transfer to neurons through these gap junctions can prevent SCI [30].

### 2.2. The Artificial Mitochondrial Transfer Pathway

Isolated mitochondria from exogenous sources have been successfully transplanted into different recipient cells in multiple in vitro and in vivo models. Exogenous mitochondria can be incorporated into recipient cells by direct injection, co-incubation, liposomes, and cell-penetrating peptides [31] (Figure 2).

Isolation and purification of mitochondria are crucial steps in studying mitochondrial transplantation. The process of isolating mitochondria through homogenization and differential centrifugation takes approximately 90 min [32]. Preble et al. describe a method for the rapid isolation of mitochondria from mammalian biopsies using a commercial tissue dissociator and differential filtration. This method aims to reduce the time required for mitochondrial purification and improve the purity and availability of mitochondria. Firstly, tissue homogenization is performed using the tissue dissociator. Then, Bacillus protease A is added to degrade proteins and release mitochondria. Subsequently, the broken tissue is filtered through 40 μm and 10 μm mesh filters. Finally, purified mitochondria are obtained through differential velocity centrifugation. The total procedure time is less than 30 min [33].

Purified mitochondria can be spontaneously taken up by target cells during co-culture [34]. Pour et al. co-incubated mitochondria isolated from L6 skeletal cells with normal H9c2 cardiomyocytes for 24 h and found that the latter could internalize the non-autologous mitochondria. Furthermore, mitochondrial transplantation enhanced ATP production and basal respiration and improved metabolism in the cardiomyocytes within a short period of time [35]. Bhattacharya et al. isolated myofibrillar interstitial mitochondria from mouse skeletal muscle and incubated them with damaged fibroblasts harboring mitochondrial DNA (mtDNA) mutations. The purified mitochondria were incorporated into the host cells, thereby enhancing mitochondrial dynamics and metabolism and restoring recipient cell function [36].

Microinjection refers to the delivery of isolated mitochondria by direct injection into the target area or through vascular infusion [37,38]. The first clinical use of mitochondrial transplantation was in pediatric patients with myocardial ischemia-reperfusion injury following coronary artery occlusion and hemodialysis. Doulamis et al. performed mitochondrial transplantation by intrarenal injection of autologous mitochondria after renal ischemia-reperfusion injury. The mitochondria were taken up by the tubular epithelium in both the cortex and medulla, protecting the kidney from ischemia-reperfusion damage, significantly improving renal function, and reducing kidney injury [39]. In addition, Mobarak et al. improved pregnancy outcomes in women with high reproductive senescence by microinjection of autologous mitochondria [40]. Although microinjection can effectively enhance transplantation efficiency, it requires considerable technical skills, as well as specialized equipment for isolating and preserving mitochondria, which limits its clinical application at present.

Liposomes are artificial lipid bilayer vesicles that are routinely used for drug delivery due to their versatility and biocompatibility and are ideal carriers for mitochondrial transfer as well. The process involves the extraction of mitochondria from donor cells and their encapsulation in liposomes, followed by the delivery of these liposomes to recipient cells. Liposomal encapsulation offers several advantages, including protection of mitochondria from the harsh extracellular environment, controlled and targeted delivery, and the possibility of site-specific release. Liposome-mediated mitochondrial transplantation has been greatly improved in terms of precision, efficiency, and functional integrity and has been successfully applied in animal models. Shi et al. transferred Mito Tracker RedH CMXRos-labeled live mitochondria to fibroblasts in NIH/3T3 mice using synthetic liposomes and verified mitochondrial transfer by fluorescence microscopy [41,42].

Peptide-mediated mitochondrial delivery (PMD) has significantly higher transfer efficiency compared to co-culture methods since it is not limited by endocytosis [43]. A mitochondria-Pep-1 complex was synthesized by incubating free mitochondria with the cell membrane-penetrating peptide Pep-1 and delivered into MERRF cells (MitoB2) and mtDNA-depleted Rho-zero cells (Mito) with transfer efficiencies of 77.48% and 82.96%, respectively. Mitochondrial transfer led to a significant reduction in the levels of mitofusin-2 (MFN2) and dynamin-related proteins and an increase in optic atrophy 1 and MFN2 in MitoB2 cells, indicating that PMD is a potential therapeutic intervention for mitochondrial diseases [44].

## 3. Mitochondrial Transplantation for Mitochondria-Deficient Diseases

### 3.1. Mitochondrial Transplantation Protects against Neuronal Damage Caused by Cerebral Ischemia

Cerebral ischemia or stroke is a condition caused by insufficient blood supply to the brain, resulting in hypoxia and damage to brain tissue. Severe cerebral ischemia or stroke can lead to neuronal damage and transient ischemic attacks [45]. Glutamate is a neurotransmitter in the central nervous system that mediates rapid excitatory synaptic responses upon binding to N-methyl-D-aspartate (NMDA)-type receptors on neuronal membranes [46]. High concentrations of glutamate have been detected in the cerebral cortex, hippocampus, and amygdala during cerebral ischemia and can cause severe neurotoxicity. NMDA receptor is a ligand-gated ion channel that is highly permeable to calcium ions. Glutamate binding effectively opens the Ca^2+^ channel [47], leading to an increase in intraneuronal Ca^2+^ concentrations [48]. The Ca^2+^ overload triggers the opening of the mitochondrial permeability transition pore (MPTP) [37], which prevents oxidative phosphorylation and ATP production, leading to mitochondrial membrane depolarization, ATP hydrolysis [49], and the release of NAD+ and Ca^2+^. The ensuing mitochondrial swelling and rupture releases ROS, cytochrome c, and other apoptosis-inducing factors [50,51]. The excessive amount of ROS released into the cytoplasm can also trigger ROS-induced ROS release (RIRR) in the neighboring mitochondria, leading to a vicious cycle of MPTP opening and continuous increase in ROS levels that ultimately leads to mitochondrial damage and cell death [52]. Transplantation of functionally normal mitochondria into injured neurons can reduce ROS production and restore ATP production, thus providing the cells with sufficient energy to activate mitochondria-targeted autophagy and enable repair [12]. Pourmohammadi-Bejarpasi et al. replicated an adult rat model of cerebral ischemia using nylon threads to block cerebral arteries and injected hucMSC-derived normal mitochondria directly into the brain using an ICV device. Examination of the brain tissue showed that the injected mitochondria were internalized into the neurons and astrocytes at the ischemic site, which was accompanied by a reduction in coagulative necrosis and restoration of normal cellular structure in the brain. In addition, the infarcted area, blood creatine phosphokinase levels, number of apoptotic cells and astrocytes, and microglial activation showed a significant decrease, resulting in improved motor function and coordination [53,54]. Existing studies have shown that Miro1 appears to be a key participant in mitochondrial transfer. In an epithelial cell injury model, Miro1 has the ability to regulate the intercellular movement of mitochondria from MSCs to epithelial cells (ECs). The overexpression of Miro1 enhances mitochondrial transfer, effectively reversing mitochondrial dysfunction in ECs and rescuing them [18]. Conversely, the knockdown of Miro1 leads to a loss of therapeutic effect. Tseng et al. conducted in vitro found that the transfer of mitochondria to oxidatively damaged neurons can improve neuronal preservation after an ischemic stroke and enhances neuronal metabolism. The research results revealed a decrease in neuronal viability and the presence of significant mitochondrial dysfunction following oxidative damage in vitro. However, co-cultivation with MCSs can restore mitochondrial function and significantly improve neuronal metabolic activity, including mitochondrial respiration and ATP production. In this study, ischemia damages neuronal mitochondria and triggers an inflammatory response, resulting in an increased production of Miro1. This, in turn, promotes mitochondrial movement and the transfer of healthy mitochondria from MCSs to neurons, potentially safeguarding neurons from apoptosis [55]. These findings suggest that mitochondrial transplantation can effectively protect against acute cerebral ischemia.

Studies show that mitochondria can be transferred from astrocytes to neurons via CD38 and cyclic ADP-ribose (cADPR) signaling. SiRNA-mediated CD38 knockdown significantly reduced the number of mitochondria released from astrocytes as well as the number transferred into neuronal cells, which mitigated the protective effect against cerebral ischemia [56]. In addition, Li et al. demonstrated the protective effects of mitochondrial transfer and internalization against severe spinal cord injury [30]. Transcellular transfer of mitochondria opens up new avenues for treating diseases of the central nervous system as well as the peripheral nervous system [57]. Several studies have shown that mitochondrial transplantation protects against neuronal damage caused by cerebral ischemia and promotes the repair of injured cardiomyocytes [58] and lung epithelial cells [26].

### 3.2. Mitochondrial Transplantation for Myocardial Ischemia-Reperfusion Injury

The heart is an oxygen-demanding organ that requires a continuous supply of energy. Unsurprisingly, mitochondria account for approximately 30% of the cardiomyocyte volume, and the functional status of these mitochondria directly influences the fate of cardiomyocytes. During ischemia, reduced blood flow to the heart limits the delivery of oxygen and nutrients, leading to mitochondrial dysfunction and impaired ATP production. This energy deficit impairs cardiac muscle contraction and triggers a series of events that culminate in myocardial ischemia-reperfusion injury (IRI) [59]. Ischemic injury, in turn, disrupts the mitochondrial inner membrane and expands the mitochondrial matrix. Therefore, the replacement of injured mitochondria with intact functional mitochondria isolated from healthy cells or tissues is a promising therapeutic strategy against myocardial IRI [18]. The transplanted mitochondria can alleviate myocardial injury by restoring lipid and glucose metabolic pathways and generating sufficient energy for cardiac functions [17].

Sun et al. constructed PEP-TPP mitochondrial complexes through the ischemic sensitivity of PEP and the mitochondrial targeting ability of TPP^+^. These complexes were able to enter ischemia-damaged cardiomyocytes through direct internalization or via endothelial cells and enhanced the respiratory capacity and mechanical contractility of cardiomyocytes, decreased the levels of pro-inflammatory cytokines such as IL-2, and reduced cardiomyocyte apoptosis after transplantation. In a mouse model of infrared radiation injury, 7.5–10 × 10^4^ intravenously injected PEP-TPP mitochondrial complexes promoted intraventricular mitochondrial engraftment in the ischemic myocardium, which significantly reduced the myocardial infarct area and provided long-term (2–4 weeks) protection against cardiomyocyte reperfusion injury [60]. The therapeutic effects of mitochondrial transplantation have also been demonstrated in rabbit and porcine models of cardiac IRI. McCully et al. isolated fresh, intact, viable, and respiring mitochondria from non-ischemic hearts. During the early reperfusion period, they injected these mitochondria into the ischemic zone to limit the damage caused by decreased mitochondrial function during ischemia. The study found that exogenous ATP and ADP were unable to protect the heart, while mitochondrial transplantation significantly reduced myocardial necrosis and cardiomyocyte apoptosis, markedly decreased infarct size (IS), caspase-3-like activity, TUNEL, as well as the release of creatine kinase isoenzymes (CK-MB) and cardiac troponin I (cTnI), alleviating myocardial injury and significantly enhancing regional and overall myocardial function recovery after ischemia [32]. Furthermore, autologous mitochondrial transplantation has unique therapeutic potential for improving ischemia-reperfusion injury and enhancing myocardial function. Isolation and preparation of autologous mitochondria from the patient’s body can prevent inflammation and rejection reactions. Masuzawa et al. observed that autologous mitochondrial transplantation led to the internalization of mitochondria by cardiomyocytes, followed by enhanced ATP production and enrichment in the generation of differentially expressed proteins associated with mitochondrial pathways and proteins responsible for the production of precursor metabolites related to energy and cellular respiration. Ultimately, this improved myocardial injury and enhances regional function [61].

Furthermore, the transplanted mitochondria in the ischemic regions of the heart exhibit intra- and extracellular functionality [62]. In one study, clusters of transplanted mitochondria were detected around the endogenous damaged mitochondria and in the vicinity of the nuclei of cardiomyocytes one hour after injection, and the damaged cells were protected by increasing oxygen consumption and ATP synthesis [61].

Apart from enhancing the energy supply to the injured myocardium, mitochondrial transplantation also initiates angiogenic, immunomodulatory, anti-apoptotic, anti-oxidant, and anti-inflammatory effects. In addition, the transplanted mitochondria can increase the circulating levels of epidermal growth factor (EGF), growth-regulated oncogenes (GRO), interleukin 6 (IL-6), monocyte chemotaxis protein 3 (MCP-3), and nuclear factor erythroid2-related factor 2 (Nrf2), which are related to the recovery of myocardial function [63,64]. During myocardial ischemia/reperfusion, EGF maintains the integrity of the myocardial endothelium by stimulating cardiomyocyte growth, proliferation, and migration [65,66]. After myocardial infarction, GRO and IL-6 promote vascularization and prevent apoptosis in the infarcted area. Along with MPC-3, these cytokines can rapidly improve cardiac remodeling after myocardial infarction through non-cardiomyocyte regenerative pathways [67]. Nrf2 activation also attenuates myocardial infarct size and preserves cardiac function through coordinated upregulation of anti-oxidant, anti-inflammatory, and autophagic mechanisms [57]. In addition, the internalized normal mtDNA can replace damaged mtDNA and exert a cardio-protective effect at the genetic level [37,68].

Mitochondrial transplantation was clinically tested for the first time in 2016 in pediatric patients with myocardial IRI. Five patients experiencing extracorporeal membrane pulmonary oxygenation (ECMO) were treated with local injections of autologous normal mitochondria isolated from the rectus abdominis muscle [69]. The systolic function improved significantly in all five patients, including four children with ischemia-induced coronary artery obstruction and one child with subepicardial ischemia-induced left ventricular hypertrophy [58]. Taken together, mitochondrial transplantation can improve myocardial function in animal models and humans and is a promising therapeutic strategy against cardiac IRI. It can be used alone or in combination with other clinical interventions, or as an adjunct to other clinical interventions. Given the availability of simple and rapid techniques for high-purity mitochondrial isolation, this approach can be applied on a large scale.

### 3.3. Mitochondrial Transplantation for Tumor Therapy

Mitochondrial dysfunction triggers the release of various death factors such as ROS, Ca^2+^, and cytochrome c, resulting in oxidative stress [70] and cellular damage [71]. Aberrant mitochondrial function and ROS overproduction in the tumor cells trigger mutations in mtDNA and nuclear DNA, leading to impaired oxidative phosphorylation that exacerbates ROS production and creates a vicious cycle [72]. The altered redox balance activates signaling pathways involved in cell survival, proliferation, and angiogenesis, further promoting tumor growth [73]. Cancer cells undergo metabolic reprogramming with an increase in glycolysis rates, a phenomenon known as the Warburg effect [74,75,76]. This transformation not only allows cancer cells to meet the energy needs for rapid proliferation [7] but also underlies radio-resistance and chemoresistance in tumors. Therefore, inhibiting glycolysis in tumor cells can sensitize them to radiation or chemotherapeutic drugs and overcome treatment resistance [77,78]. Mitochondrial transplantation in tumor cells can reduce aerobic glycolysis, block cell cycle progression by downregulating cycle-related proteins, and activate the intrinsic apoptosis pathway by upregulating pro-apoptotic proteins, eventually inhibiting cell proliferation [79]. Sun et al. found that endocytosis-mediated mitochondrial transplantation from normal human astrocytes to glioma cells rescued aerobic respiration, attenuated the Warburg effect, and improved radiosensitivity of gliomas. Furthermore, endocytosis of mitochondria into the glioma cells was mediated by nicotinamide adenine dinucleotide (NAD^+^)-CD38-cADPR-Ca^2+^ signaling [80]. CD38 is a single-chain type I transmembrane glycoprotein that catalyzes the generation of cADPR from NAD^+^ and transports cADPR into the cell in the form of a homodimer [81]. cADPR acts as a second messenger in the intracellular signaling cascade that mediates the release of intracytoplasmic Ca^2+^ and regulates changes in the cytoskeleton, endocytosis, or exocytosis, which may be responsible for transcellular mitochondrial transfer [82,83].

Elliott et al. co-cultured the mitochondria isolated from normal breast epithelial cells (MCF-12A) with breast cancer cell lines (MCF-7, MDA-MB-231, and NCI/ADR-Res). The introduction of normal mitochondria into the breast cancer cells inhibited proliferation and enhanced their sensitivity to doxorubicin, abraxane, and carboplatin [84]. Recent studies have shown that mitochondrial transplantation can also inhibit the proliferation of melanoma cells and induce apoptosis. Chang et al. isolated normal and *A8344G*-mutated mitochondria from homeoplasmic 143B osteosarcoma cells and delivered them to MCF-7 breast cancer cells through passive uptake or Pep-1. Mitochondrial transplantation induced apoptosis in the recipient cells by increasing nuclear translocation of apoptosis-inducing factor AIF [85]. Yu et al. administered intact mitochondria extracted from mouse livers into mice harboring subcutaneous and metastatic melanomas via the intravenous route. Transplantation of the healthy mitochondria induced cell cycle arrest and apoptosis by downregulating transcription of the anti-apoptotic protein BCL-2 and upregulating the mitochondria-associated apoptosis-inducing factor gene (*Aifm3*) transcripts. In addition, autophagy-related proteins such as LC3 were also upregulated at the transcriptional level. Finally, mitochondrial transplantation induced transcriptional silencing of proliferation-related and anti-apoptotic genes via histone methylation [86].

Hypoxia is one of the recognized hallmarks of cancer and contributes to the resistance of tumor cells to chemotherapy and radiotherapy. High glycolysis rates in hypoxic tumor cells support rapid tumor cell proliferation, and the metabolites create an acidic environment that is conducive to tumor growth [87,88]. Spees et al. treated A549 cells with ethidium bromide to induce mtDNA mutations and depletion and inhibit aerobic respiration. Following co-incubation of the mtDNA-depleted (A549ρ0) cells with human bone marrow-derived skin fibroblasts, the latter formed cytoplasmic extensions toward the target cells. Mitochondrial transplantation from the fibroblasts via these extensions restored oxidative phosphorylation in A549ρ0 cells, decreased the level of oxygen deprivation in the tumor cells, and attenuated the degree of malignancy [89].

In conclusion, mitochondrial transplantation has shown significant potential in the field of tumor therapy. Transplantation of healthy mitochondria into cancer cells can inhibit cell proliferation, enhance sensitivity to chemotherapy and radiation, and induce apoptosis. In addition, mitochondrial transplantation has been shown to rescue aerobic respiration and attenuate the Warburg effect in glioma and breast cancer cells, improve their radiosensitivity [80,90], induce apoptosis in melanoma cells, and inhibit lung cancer cells under hypoxic conditions [86,89]. These findings highlight the potential of mitochondrial transplantation as a novel tumor treatment strategy.

## 4. Discussion

In this review, we have summarized the recent research on the therapeutic potential of mitochondrial transplantation in diseases related to defective mitochondria. Neurodegenerative disorders such as Alzheimer’s disease, Parkinson’s disease, aging-related neuropathy, type 2 diabetes, and cancer are associated with mitochondrial dysfunction [91,92,93,94]. Mitochondrial transplantation has been applied in the therapeutic management of neurodegenerative disorders due to its ability to restore or enhance normal mitochondrial functionality and energy supply, thereby exerting neuroprotective effects. In addition, mitochondrial therapy for cardiac ischemia-reperfusion has progressed to clinical testing. Intact mitochondria can rescue target cells in injured tissues by restoring normal physiological functions, maintaining cellular homeostasis, and replacing the damaged mitochondria through fusion and fission, which remodels the mitochondrial network and promotes cell survival [95].

Mitochondrial dysfunction in tumor cells is closely related to tumorigenesis and progression, and studies have revealed multifaceted roles of mitochondrial dynamics, metabolic reprogramming, and signaling pathways in tumorigenesis. Furthermore, selective targeting of mitochondria in cancer cells can overcome radioresistance and chemoresistance by restoring mitochondrial function [96]. The efficacy of mitochondrial transplantation combined with radiotherapy and chemotherapy against resistant cancer cells will have to be evaluated through animal models and clinical trials.

Unfortunately, there is currently no effective method for the long-term storage of mitochondria, which have to be used immediately after extraction. The isolated mitochondria are prone to inner and outer membrane damage, resulting in greatly reduced function and activity [97]. Therefore, it is crucial to establish methods for optimal mitochondrial isolation, quality control, storage, and transplantation in order to preserve intact and viable mitochondria for clinical applications.

Monogenic mitochondrial diseases are caused by specific gene mutations that affect mitochondrial function, such as mitochondrial encephalopathy, lactic acidosis, and stroke-like episodes (MELAS) syndrome, Leigh syndrome, and Neuropathy, ataxia, and retinitis pigmentosa (NARP) syndrome [98,99,100]. Mitochondrial transplantation has the potential to treat monogenic mitochondrial diseases by restoring normal mitochondrial function and alleviating symptoms associated with these diseases. Taivassalo et al. found that mitochondrial myopathy patients have limited ability to extract usable oxygen from the blood [101]. Due to inadequate energy for intracellular magnesium transport, intracellular hypomagnesemia may be a contributing factor to MELAS syndrome [102]. From this perspective, mitochondrial transplantation may also improve aerobic metabolism, enhance energy production, and play a role in the treatment and prevention of stroke-like episodes in MELAS syndrome. Although mitochondrial transplantation provides some hope for the targeted treatment of monogenic mitochondrial diseases, there is still a long way to go for the successful application of this method in laboratory experiments and clinical practice.

Autologous mitochondrial transplantation is a promising therapeutic intervention for various diseases. Since mitochondria are processed outside the body and reintroduced into the same patient, this therapy would be of high specificity and would eliminate post-transplantation immune rejection. From the perspective of precision medicine, mitochondrial transplantation can be effective against different pathological processes. We also believe that every patient with mitochondrial defective diseases will possess their own mitochondria biological agents in the near future.

## Figures and Tables

**Figure 1 ijms-25-01175-f001:**
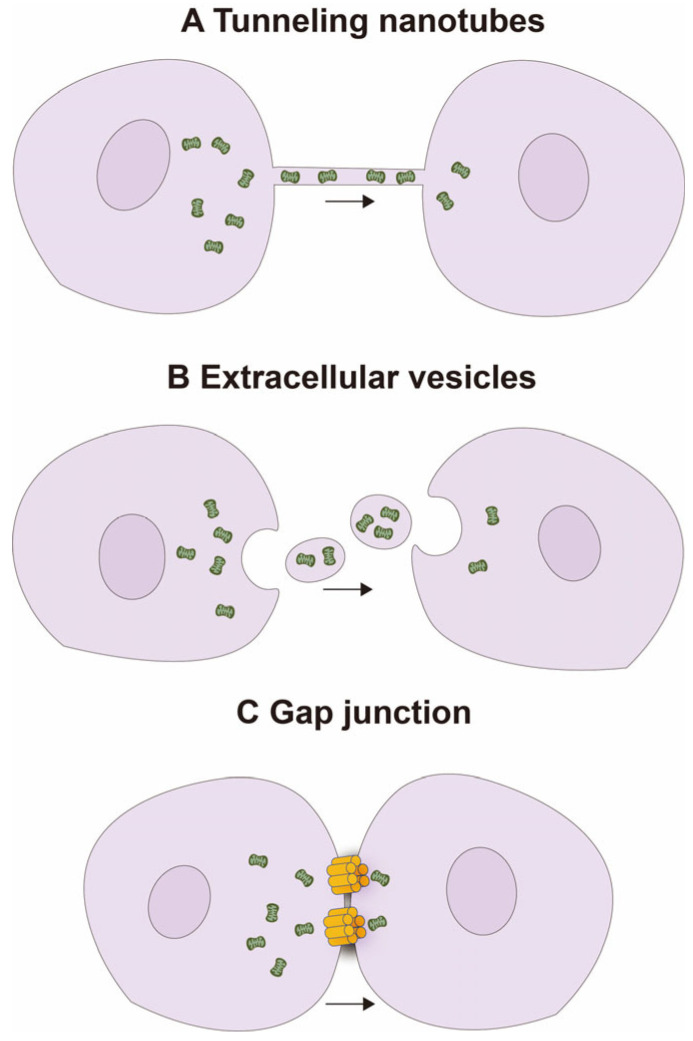
Natural mitochondrial transfer mechanisms.

**Figure 2 ijms-25-01175-f002:**
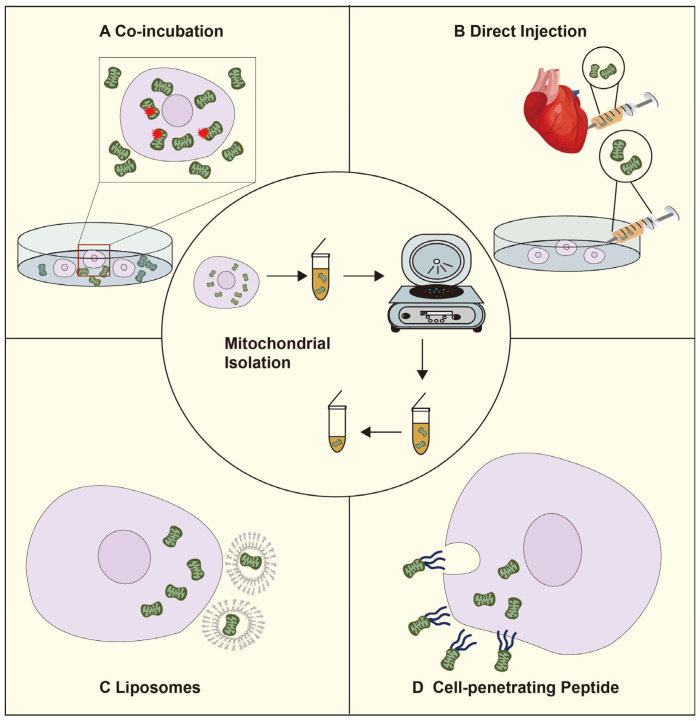
The artificial mitochondrial transfer pathway.

## Data Availability

Not applicable.

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
