# Peer review of "The Research Progress of Mitochondrial Transplantation in the Treatment of Mitochondrial Defective Diseases"

_ijms, 2024, doi:10.3390/ijms25021175_

Round 1

Reviewer 1 Report

Comments and Suggestions for Authors

This article by Hu et al. discusses the state of the art in the use of mitochondria for the treatment of diseases, more specifically, mitochondrial transplantation. 

Although the work is well focused, in my view it lacks depth in some experimental aspects. 

For example, in Figure 2, in all cases isolated mitochondria are used for transplantation. However, the text does not discuss how this purification and isolation of mitochondria takes place. Is this process experimentally the same in each of the techniques used? I think that, if in all cases, the first step is the purification of the mitochondria, there should be a paragraph about this procedure. 

In point 3.3, from line 299 to 302, that paragraph seems to belong to some kind of template for writing that content. Is it so?

Typos: 

Line 17: review instead of re-view

Reviewer 2 Report

Comments and Suggestions for Authors

The authors have prepared a great and well-written review disclosing recent progress of mitochondrial transplantation in the treatment of mitochondrial defective diseases. This manuscript may be tremendously interesting for the community. I only have a few remarks:

-                L.17 : «In this re-view» probably should be corrected to «In this review»

-                L.121 «transplant-ed » should be corrected to «transplanted»

-                L. 152 «con-trolled»  should be corrected to «controlled»

-                L. 176 «Gluta-mate » should be corrected to «Glutamate»

-                The tunneling nanotubes (TNTs) and mesenchymal stem cells (MSCs) acronyms have been introduced multiple times throughout the article. Some abbreviations are not used after being introduced. Some abbreviations are introduced not from the first mention of the term.

-                Mentationing the acronym CTG-TNTs (obviously referring to CellTracker Green) does not bring helpful information but complicates the understanding. It rather reflects technical details then crucial findings in the cited article so I would not use it.

-                L. 140-141 : «Sitaram et al. successfully delivered mitochondria to the ischemic region of the heart by direct microinjection and restored myocardial function». Factually, the cited work does not provide evidence that mitochondria were delivered successfully. The authors only showed clinical outcomes following the myocardial injections of isolated mitochondria.

-                L.169-171 : «Cerebral ischemia is an acute and chronic neurodegenerative disease caused by in- sufficient blood supply, which can lead to neuronal damage, transient ischemic attack, aneurysm and stroke [43].». Cerebral ischemia is not considered a neurodegenerative disease, and it scarcely may lead to aneurysm. Moreover there are no causative relationships between Cerebral ischemia and stroke/TIA, because in some sense those conditions are synonimous.

-                L. 204 «enhance» probably should be corrected to «enhances»

-                L299-302. The sentence seems to be not on the right place.

-                The rationale of mitochondrial transplantation for the treatment of tumors seems a bit looks contradictory to the previous sections. For instance, initiating of apoptosis following mitochondrial transplantation is not a desirable event in the treatment of cardiac or cerebral ischemia. Such a contradiction should be emphasized more clearly.

-                I would recommend to more in details discuss possible applications of the mitochondrial transplantation for the treatment of monogenic mitochondrial diseases, not only mitochondrial dysfunction in general.
